# OpenReview forum: "RLCD: Reinforcement Learning from Contrastive Distillation for LM Alignment"
_ICLR.cc/2024/Conference — ICLR 2024 poster_

### Official Review · Reviewer_bkDY · 2023-10-31

**Soundness:** 2 fair
**Presentation:** 3 good
**Contribution:** 2 fair
**Rating:** 6
**Confidence:** 3

**Summary:**

This paper proposes Reinforcement Learning from Contrastive Distillation, which uses an unaligned language model to produce contrasting response pairs from both positive and negative prompts in terms of a desired attribute, e.g. harmlessness or helpfulness. It is argued that responses generated in this way are more distinguishable, producing a better signal-to-noise ratio. The produced rankings between pairs are then used for training a preference model, and subsequently, RLHF. Experiments are conducted on three alignment tasks, showing better performance than RLAIF and context distillation baselines.

**Strengths:**

1. The paper is well-written and easy to follow.
2. The method is clearly motivated, and the authors make a detailed argument on the problems with the prior work.
3. Experiments are conducted over multiple domains, demonstrating the generality of the method.

**Weaknesses:**

1. The analysis on the preference model shows that the preference model produced by RLCD is, while better than the baseline, still not very good, especially on the harmlessness attribute (Tab. 5). It is not clear how this slight advantage over chance (2.4%~5.9%) translates into a much better downstream performance after RLHF.
2. As shown in Appendix C, RLAIF-Few-30B produces both a better preference model and a better-aligned language model than RLCD-30B on the harmlessness benchmark, which is attributed to few-shot prompting by the authors. It seems that this technique can also be integrated into RLCD to enable a fairer comparison.
3. The advantage of RLCD over RLAIF shrinks going from 7B to 30B (Tab. 2). It remains to be seen whether RLCD (or RLCD-Rescore) can scale to yet larger language models that are arguably better at differentiating responses near the decision boundary.

**Questions:**

Not atm

---

> ### Author Response · Authors · 2023-11-17
> **Reply to Reviewer bkDY**
>
> Thank you for your helpful comments! We appreciate your comments about our idea being clearly motivated and about the generality of our method across multiple domains. We address your questions below.
>
> > **Concern 1: The analysis on the preference model shows that the preference model produced by RLCD is, while better than the baseline, still not very good, especially on the harmlessness attribute (Tab. 5). It is not clear how this slight advantage over chance (2.4%~5.9%) translates into a much better downstream performance after RLHF.**
>
> Actually, we think that the preference model doesn’t need to be that far above chance to enable good downstream performance. For example, Anthropic’s original RLHF paper [1] observed only 63% agreement between their own researchers and crowdworkers on their helpful-honest-harmless labels. After all, the human labels are on pairs of model outputs generated from the same prompt, so there is often not a clear winner within a pair.
>
> We observe that harmlessness is perhaps even a bit noisier than helpfulness: we checked GPT-4’s agreement with humans and observed only 60.5% agreement on harmlessness compared to 73.0% on helpfulness. So on harmlessness, that would put e.g., 5.9% above chance as closing 56% of the gap between chance and GPT-4. So we think it’s reasonable to observe decent downstream performance with our reward models even if the advantage over chance doesn’t look like much.
>
> > **Concern 2: As shown in Appendix C, RLAIF-Few-30B produces both a better preference model and a better-aligned language model than RLCD-30B on the harmlessness benchmark, which is attributed to few-shot prompting by the authors. It seems that this technique can also be integrated into RLCD to enable a fairer comparison.**
>
> We intentionally used zero-shot prompts in our main experiments to minimize the influence of prompt selection for fair comparison (more detailed discussion under Reviewer HiRS, Concern 1); adding few-shot examples would introduce an additional variable. We mainly included RLAIF-Few-30B in Appendix C to check that we can replicate the results from [2], to make sure that RLAIF’s often poor performance when we do not use both (1) few-shot and (2) 30B scale is not due to implementation differences.
>
> To be clear, we think the final language model learned by RLAIF-Few-30B is not good - as shown in Table 17 in Appendix C, the model distribution collapses to a generic but unhelpful output, with *nearly all* outputs being a variation of “I'm sorry, but I'm not sure how I can help with that. Can I ask some questions to help me understand your problem better?”. Quantitatively, the fraction of distinct 3-grams (Dist-3) within a sample of 10000 tokens is only 18% (!) for RLAIF-Few-30B, compared to over 90% for most other experiments (see new Appendix J.3; we’ve also added more discussion on the mode collapse to Appendix C).
>
> In any case, we’ve now run RLCD-Few-30B on harmlessness too. RLCD-Few-30B also collapses to a single generic output: “I'm here to help you get the information you need. Can I help you today?”. GPT-4 happens to prefer RLAIF-Few-30B’s generic output, but we think that it’s more indicative to look at the preference model’s performance since both models are so heavily mode-collapsed. RLCD-Few-30B’s preference model’s accuracy for agreeing with humans is 62.8% (avg. prob of agreement = 0.599), a good bit higher than the 57.0% for RLAIF-Few-30B’s (avg. prob of agreement = 0.540).
>
> > **Concern 1: The advantage of RLCD over RLAIF shrinks going from 7B to 30B (Tab. 2). It remains to be seen whether RLCD (or RLCD-Rescore) can scale to yet larger language models that are arguably better at differentiating responses near the decision boundary.**
>
> We totally agree with this point, as touched upon in our Discussion and our Limitations sections, and we’d be happy to emphasize it more for clarity. However, even if it turns out that RLCD in its current form is not as useful as model scale exceeds 30B (which is still unclear), we believe that we have still made valuable research contributions, as described in “Additional Aspects of Our Contribution” under the General Response.
>
> [1] Bai, Yuntao, et al. "Training a helpful and harmless assistant with reinforcement learning from human feedback." arXiv preprint arXiv:2204.05862 (2022).
>
> [2] Bai, Yuntao, et al. "Constitutional ai: Harmlessness from ai feedback." https://arxiv.org/abs/2212.08073

---

> > ### Comment · Reviewer_bkDY · 2023-11-22
> > **Thanks for your rebuttal**
> >
> > Hi,
> >
> > Thanks for your time spent on the rebuttal. I appreciate your honesty. Most of my concerns have been addressed and there are lots of evaluations added in. I agree to raise my review score.

---

> > > ### Author Response · Authors · 2023-11-22
> > > **Thanks!**
> > >
> > > Thank you again for your careful review, and we really appreciate you raising your score!

---

### Official Review · Reviewer_Cuc6 · 2023-11-01

**Soundness:** 2 fair
**Presentation:** 3 good
**Contribution:** 2 fair
**Rating:** 5
**Confidence:** 3

**Summary:**

This paper proposes a method called RLCD that incorporates the idea of context distillation into RLHF framework. Instead of using a single prompt to elicit preference data, RLCR does this by constructing two manually augmented prompts of positive and negative instructions. The authors show the effectiveness of RLCR by comparing it with RLAIF and SFT context distillation on a dataset.

**Strengths:**

1.The idea is interesting and the motivation seems clear.

2.The paper is well-written.

**Weaknesses:**

1.The proposed method is too simple that seems like a prompting trick in preference data construction.

2.More comprehensive methodology and solid experiments (e.g., stronger baselines and deeper analyses) are needed to improve the contribution and soundness of this paper.

**Questions:**

1. Will the proposed method reduce the diversity of preference data?

2. What if the original prompt already contains positive or negative instructions?

---

> ### Author Response · Authors · 2023-11-17
> **Reply to Reviewer Cuc6**
>
> Thank you for your helpful comments! We appreciate your comments that our idea is interesting and well-motivated, and that our paper is well written. We address your questions below.
>
> > **Concern 1: The proposed method is too simple that seems like a prompting trick in preference data construction.**
>
> The core idea of RLCD is to change the generation process of the preference outputs, moving away from noisy near-boundary examples to more reliable far-from-boundary examples; this tradeoff is both empirically effective and automatable. Please see “Additional Aspects of Our Contribution” under the General Response for a more detailed discussion of our contributions; we would also like to highlight our theoretical motivation for RLCD in the new Appendix N, which we summarize in “Theoretical Motivation for RLCD” in the General Response.
>
> RLCD then relies on prompting because—given that we want to change the generation process of the preference outputs—prompting seems to be the simplest implementation of our idea. We view this simplicity of implementation as a strength which makes RLCD easier to reuse or adapt in new applications; we note that reviewers HiRS, UFW8, and RAFU mentioned simplicity / straightforwardness of implementation of RLCD under Strengths as well.
>
>
> > **Concern 2: More comprehensive methodology and solid experiments (e.g., stronger baselines and deeper analyses) are needed to improve the contribution and soundness of this paper.**
>
> We’ve now added experiments according to your questions. Was there something else in particular you were looking for?
>
> > **Question 1: Will the proposed method reduce the diversity of preference data?**
>
> We’ve now checked the preference data diversity by measuring the percentage of distinct 1-, 2-, and 3-grams among 10K tokens of preference data for both RLAIF and RLCD (see new Appendix K.2). The metrics for RLCD’s preference data are overall very similar to those of RLAIF; if anything RLCD has higher diversity more often than not.
>
>
>
> > **Question 2: What if the original prompt already contains positive or negative instructions?**
>
> We think adding positive or negative instructions to the original prompt is fundamentally orthogonal to the RLCD idea, which is mainly about creating *contrast* in the outputs $o_+$ and $o_-$ by using *contrasting* prompts $p_+$ and $p_-$. To confirm, we’ve also added an experiment in the new Appendix L, where we observe that adding positive instructions to the prompt doesn’t change RLAIF’s performance compared to RLCD.

---

> ### Comment · Reviewer_Cuc6 · 2023-11-21
>
> The author indeed addresses the issues I was concerned about. However, I argue that the core innovativeness of the method (using LLM to generate pairwise preference data) is still insufficient for publication; it seems more like a variant of context distillation. Considering this work's extensive experimental validation and case analysis, I raise the score to 5.

---

> > ### Author Response · Authors · 2023-11-21
> > **Thanks!**
> >
> > Thank you again for your helpful review, and we are glad that we addressed your concerns.
> >
> > While context distillation is an important inspiration for our method, and some prompts may be similar in implementation, we emphasize that the underlying goal of RLCD---building higher-contrast preference pairs---is meaningfully different; see for example our new analysis in Appendix N. Some additional details are that context distillation cannot use a negative prompt $p_-$ like RLCD, and empirically performs much worse than RLCD.
> >
> > We hope this can help address your concern about our contributions; please also let us know if you have anything else to ask. Either way, we sincerely appreciate you raising your score.

---

### Official Review · Reviewer_RAFU · 2023-11-02

**Soundness:** 3 good
**Presentation:** 3 good
**Contribution:** 2 fair
**Rating:** 6
**Confidence:** 3

**Summary:**

The paper introduces Reinforcement Learning from Contrastive Distillation (RLCD), a novel method to align language models with human values without relying on human feedback. It's designed to overcome limitations in previous approaches like Reinforcement Learning from AI Feedback (RLAIF) and context distillation. RLCD operates by generating two contrasting model outputs using positive and negative prompts, with the positive prompts encouraging adherence to desired principles (e.g., harmlessness) and negative prompts doing the opposite. This method creates clearer preference pairs for training a preference model, which is then used to improve a base unaligned language model via reinforcement learning. The paper demonstrates that RLCD outperforms existing methods across various tasks and model scales, confirming its effectiveness in aligning language models more closely with desired human values.

**Strengths:**

- The paper is well written and easy to read
- The proposed RLCD method seems straightforward to implement and can be generalised
- By generating clearer preference pairs without human annotations, RLCD reduces the cost and time associated with collecting high-quality human preference data.
- The paper provides empirical evidence that RLCD outperforms existing methods across different tasks and scales.

**Weaknesses:**

- The GPT-4 evaluated the baseline RLAIF outperforming the proposed RLCD method for the 30B model size.
- The contribution of the paper is limited

**Questions:**

In the GPT-4 evaluation results (Table 3), the authors mentioned that "The gap between RLCD and all baselines is especially large when using LLaMA-7B for preference data simulation.". Indeed, comparing row 2 and row 5: the RLCD-7B has a larger advantage over RLAIF-7B compared with RLCD-30B vs. RLAIF-30B. Similarly for the human evaluation in Table 2.
And in the Helpfulness and Outlining tasks, the baseline RLAIF-30B scored higher than the proposed RLCD-30B). I have a few questions on this:
- Why does the proposed RLCD algorithm gain a bigger advantage on the smaller LlaMA model when compared with its baselines? Is it due to the poor performance of RLAIF with smaller models?
- For the helpfulness and outlining tasks, the baseline RLAIF-30B outperformed the proposed RLCD. What is the cause of this result and should we generally adopt RLAIF when a larger LLM is available?
- The human evaluation always preferred RLCD compared with RLAIF, in contrast to GPT 4's preference on RLCD-30B. Can the authors please provide some insights into the cause of the difference in the GPT4 vs. human evaluation?

The downstream fine-tuning is using PPO. Although not the main contribution of the paper, can the authors please provide the details of the downstream fine-tuning procedures? For example, what are input and output of the RL model, what is the reward used by PPO derived from the upstream preference generation? And the overall process of the RL fine-tuning? This information would be useful for a general audience who would like to use the proposed RLCD method.

---

> ### Author Response · Authors · 2023-11-17
> **Reply to Reviewer RAFU**
>
> Thank you for your helpful comments! We appreciate your comments about the implementation ease and generalizability of our method and its empirical effectiveness. We address your questions below.
>
> > **Concern 1: The GPT-4 evaluated the baseline RLAIF outperforming the proposed RLCD method for the 30B model size.**
>
> Please see Question 1b below.
>
> > **Concern 2: The contribution of the paper is limited**
>
> Please see “Additional Aspects of Our Contribution” under the General Response.
>
> > **Question 1: In the GPT-4 evaluation results (Table 3), the authors mentioned that "The gap between RLCD and all baselines is especially large when using LLaMA-7B for preference data simulation.". Indeed, comparing row 2 and row 5: the RLCD-7B has a larger advantage over RLAIF-7B compared with RLCD-30B vs. RLAIF-30B. Similarly for the human evaluation in Table 2. And in the Helpfulness and Outlining tasks, the baseline RLAIF-30B scored higher than the proposed RLCD-30B). I have a few questions on this:**
>
> > **Question 1a: Why does the proposed RLCD algorithm gain a bigger advantage on the smaller LlaMA model when compared with its baselines? Is it due to the poor performance of RLAIF with smaller models?**
>
> Indeed, RLAIF works quite poorly with smaller models, which struggle to label preference data accurately using the RLAIF scoring prompts (see Tables 24 and 25 for examples, if interested). Please see our “Theoretical Motivation for RLCD” under the General Response, which includes potential explanations for why RLCD works so much better on 7B, or the new Appendix N for complete details on these theoretical intuitions.
>
> > **Question 1b: For the helpfulness and outlining tasks, the baseline RLAIF-30B outperformed the proposed RLCD. What is the cause of this result and should we generally adopt RLAIF when a larger LLM is available?**
>
> To be clear, we trust the human evaluations more; see Question 1c below for discussion of human-GPT4 disagreements. Nevertheless, you raise an important question of whether we should use RLCD with even larger LLMs, as the difference between RLCD and RLAIF is much smaller at 30B compared to 7B. Though we’ve already tested up to 30B scale, it’s indeed not yet clear whether RLCD will scale to even larger model sizes (as we touched upon in our Discussion and our Limitations sections too).
>
> However, even without considering whether RLCD in its current form scales well to >30B scale, we believe that we have still made useful research contributions; for instance, please see “Additional Aspects of Our Contribution” under the General Response.
>
> > **Question 1c: The human evaluation always preferred RLCD compared with RLAIF, in contrast to GPT 4's preference on RLCD-30B. Can the authors please provide some insights into the cause of the difference in the GPT4 vs. human evaluation?**
>
> We discussed the annotation disagreements briefly in Appendix G, where we observed quantitatively that GPT-4’s agreement with humans is lower in two main cases.
>
> First, when evaluating helpfulness on the harmlessness prompt set, GPT-4’s own harmlessness alignment often prevents it from saying which answer is more helpful.
>
> Second, specifically when comparing RLCD-30B to RLAIF-30B, the disagreement is higher because both models are actually producing quite good outputs (see e.g., example helpfulness outputs for RLCD-30B and RLAIF-30B in Table 27), so the comparison may simply be more subjective. For example, with outlines, we noticed qualitatively that RLCD’s outputs seemed a bit more surprising, which was fine for humans but somewhat dispreferred by GPT-4. So we think GPT-4 likely just subjectively prefers a particular style, and that it’s a bit more reliable to survey humans as a result.
>
> > **Question 2: The downstream fine-tuning is using PPO. Although not the main contribution of the paper, can the authors please provide the details of the downstream fine-tuning procedures? For example, what are input and output of the RL model, what is the reward used by PPO derived from the upstream preference generation? And the overall process of the RL fine-tuning? This information would be useful for a general audience who would like to use the proposed RLCD method.**
>
> Good point, we previously included the implementation details and hyperparameters, but we’ve now added a description of the actual PPO fine-tuning procedure in Section 3.1.1 in the main text.

---

> > ### Comment · Reviewer_RAFU · 2023-11-22
> > **thank the authors for their rebuttal**
> >
> > I thank the authors for their rebuttal and updating the PPO details. I have no further questions.

---

> > > ### Author Response · Authors · 2023-11-22
> > > **Thanks!**
> > >
> > > Thank you again for your careful review!

---

### Official Review · Reviewer_UFW8 · 2023-11-06

**Soundness:** 3 good
**Presentation:** 3 good
**Contribution:** 2 fair
**Rating:** 6
**Confidence:** 2

**Summary:**

This paper proposes a new method called Reinforcement Learning from Contrastive Distillation (RLCD) for aligning Language models (LMs) to follow principles expressed in natural language. RLCD utilizes contrasting prompts encouraging and discouraging adherence to principles, resulting in differentiated model outputs and cleaner preference labels, eliminating the need for human feedback. Based on the generated preference pairs, RLCD trains a preference model that captures desired behavior. The trained preference model guides a reinforcement learning process to refine an unaligned base LM, aligning it with the specified principles. RLCD outperforms existing methods like RLAIF and context distillation on diverse tasks including harmlessness, helpfulness, and story outline generation. RLCD demonstrates effectiveness with both small (7B) and large (30B) model sizes for simulating preference data. Overall, RLCD offers a novel method for human-free alignment of language models, surpassing existing techniques and demonstrating promising scalability.

**Strengths:**

RLCD is a neat idea that is simple yet effective. It requires some changes to the prompt to force the model to output more contrastive positive and negative outputs, which yield significant improvement in practice. I think the simplicity and effectiveness of the method is of value to the community.

Moreover, the empirical results seem quite strong, which makes the method more convincing.

The paper is also easy to follow and understand.

**Weaknesses:**

The method seems straightforward and is less technically strong. The main technical contribution is the simple changes on the prompt design when generating the pair of responses for the preference data. While such changes make sense intuitively, i.e. making both responses more contrastive, the authors didn't show much principled analysis on why such design can be better than direct RLAIF. I think it would be helpful to give some more technical/principled explanation on RLCD.

Moreover, I wonder if this design is important when there's human feedback data presented in the preference dataset as well, which is more common in practice. It would be interesting to see if RLCD would still make such a big difference in practice with some human feedback data in the mix.

**Questions:**

1. Please clarify the technical details of RLCD, ideally some theoretical justifications.
2. Please show some experiments in scenarios where there's human preference data available in the data mixture.

---

> ### Author Response · Authors · 2023-11-17
> **Reply to Reviewer UFW8**
>
> Thank you for your helpful comments! We appreciate your comments about the simplicity and effectiveness of our method in practice and our paper being easy to follow and understand. We address your questions below.
>
> > **Concern 1: The method seems straightforward and is less technically strong. The main technical contribution is the simple changes on the prompt design when generating the pair of responses for the preference data. While such changes make sense intuitively, i.e. making both responses more contrastive, the authors didn't show much principled analysis on why such design can be better than direct RLAIF. I think it would be helpful to give some more technical/principled explanation on RLCD.**
>
> Thanks for this excellent suggestion! We’ve now added a technical explanation for our intuitions; please see “Theoretical Motivation for RLCD” under the General Response, and the new Appendix N for complete details.
>
>
> > **Concern 2: Moreover, I wonder if this design is important when there's human feedback data presented in the preference dataset as well, which is more common in practice. It would be interesting to see if RLCD would still make such a big difference in practice with some human feedback data in the mix.**
>
> Since we have the original Anthropic human-labeled preference data for harmlessness and helpfulness, we’ve now tested harmlessness and helpfulness on 7B model scale with human preference labels mixed in (see new Appendix M). Even with as many as 20% human labels, RLCD still outperforms RLAIF by a decent margin according to GPT-4, though the difference is naturally smaller than with no human labels.
>
>
> > **Question 1: Please clarify the technical details of RLCD, ideally some theoretical justifications.**
>
> Please see “Theoretical Motivation for RLCD” under the General Response.
>
> > **Question 2: Please show some experiments in scenarios where there's human preference data available in the data mixture.**
>
> Please see response to Concern 2.

---

### Official Review · Reviewer_HiRS · 2023-11-07

**Soundness:** 3 good
**Presentation:** 3 good
**Contribution:** 2 fair
**Rating:** 6
**Confidence:** 3

**Summary:**

Reinforcement learning (RL) from human feedback has been very effective in aligning large language models (LLM) to human preferences. In particular, RLHF requires human preference data to learn a reward model for optimizing the LLM with RL. However, collecting human preference data can be very expensive. This paper proposed to simulate human preferences using ideas from context distillation. Unlike context distillation, which only modifies the original prompt to be more positive, this paper modifies the original prompt to be both positive and negative, creating a preference dataset from the new prompt-generation pairs. The author's results show that this idea empirically performs better than two competitive baselines across various tasks.

**Strengths:**

- The proposed idea is very simple, and it is intuitive why the idea should perform well in practice.
- The authors thoroughly evaluated their approach to baseline approaches using both GPT-4 and human evaluation.
- The paper is well-written, and it is easy to follow.
- The authors perform experiments on a 7B and 30B model to show how robust their proposed technique is at scale.

**Weaknesses:**

- It is hard to understand if the performance is coming from the prompts themselves or the proposed algorithm.
- The authors only report GPT-4 and human evaluation but do not report RM-score or standard NLP metrics (e.g., perplexity or output-perplexity)
- The authors do not provide a thorough description of the outlining prompts task, and there does not seem to be any references for this task, so it is very hard to understand the task's difficulty.

**Questions:**

- How did you decide on the prompt affix pairs?
- Why is having more than one prompt affix pair important?
- Given that you automatically assume $o_{+}$ is preferred - how often does $o_{+}$ have a lower reward with respect to a held-out reward function?
- Why would training examples far away from the boundary be better than training examples close to the boundary? I would assume that the points far away could be easy to classify.
- Could you elaborate on how you performed your GPT-4 evaluation? What prompt did you use? How did you shuffle the data? Etc?
- Could you provide other quantitative metrics for all algorithms considered in your experiments? (e.g., RM-score, perplexity, output-perplexity, etc.)
 - Did you perform GPT-4 evaluation using comparisons from the algorithms-generated output and human-generated data for a given prompt?
- Could you provide other diversity metrics on the outputs of the text generated? (e.g., the ratio of distinct n-grams (Distinct-1, Distinct-2), average length of sentences, or count of n-grams in the generated text [1]). The text in Table 4 implies that RLCD generations are much longer than the base model sentences.
- Could you elaborate on the RLCD-Resouce model setup? In particular, what does it mean to re-label the same scoring prompts as in RLAIF?
- For RLAIF, did you run an experiment where you sample two outputs from the $p_{+}$ positive affix prompts? This provides RLAIF algorithms with modified prompts similar to RLCD and would reduce the advantage that RLCD has to strictly have the altered $p_{-}$ prompts.

[1] A diversity-promoting objective function for neural conversation models by Li et al. 2015

---

> ### Author Response · Authors · 2023-11-17
> **Reply to Reviewer HiRS (part 1)**
>
> Thank you for your helpful comments! We appreciate your comments about the intuitiveness of our method’s usefulness in practice, the thoroughness of our experiments, and our paper being well-written. We address your questions below, including several new analyses following your suggestions.
>
> > **Concern 1: It is hard to understand if the performance is coming from the prompts themselves or the proposed algorithm.**
>
> All RLAIF scoring prompts and RLCD affix pairs are shown in Appendix A, together with a description of how we selected them to maintain fairness of comparison with our baselines.
>
> RLAIF uses “discriminative” prompts for scoring while RLCD uses the prompt affix pairs in generation, so there are necessarily some differences where the language is unnatural when converting directly from one to the other, but we attempted to match the general meaning. For example, for harmlessness, we used the original 16 scoring prompts from [1] for RLAIF, and then constructed 16 corresponding affix pairs for RLCD to paraphrase the qualities being evaluated in those scoring prompts. Due to ensembling 16 prompts, we think that our harmlessness experiments should be particularly robust to the effect of prompt selection; the results also hold up when we run a version of harmlessness with a different set of more “focused” prompts in Appendix D. For helpfulness and outlining we wrote our own scoring prompts for RLAIF, and each scoring prompt directly corresponds to an affix pair designed to encourage differences on the same axis being measured.
>
> Meanwhile, context distillation directly uses the same positive prompt $p_+$ as RLCD, to minimize the effect of prompt selection as much as possible.
>
>
>
> > **Concern 2: The authors only report GPT-4 and human evaluation but do not report RM-score or standard NLP metrics (e.g., perplexity or output-perplexity)**
>
> Thanks for the suggested metrics, we’ve now checked these.
>
> For RM-score (new Appendix J.1), we have human preference data for harmlessness and helpfulness, so we train a held-out reward model on those. RLCD’s outputs are the highest reward, except on helpfulness at 30B scale only, where the reward is slightly lower than RLAIF.
>
> For the conditional perplexity of outputs (new Appendix J.2), RLCD’s numbers are generally similar to those of baselines.
>
> > **Concern 3: The authors do not provide a thorough description of the outlining prompts task, and there does not seem to be any references for this task, so it is very hard to understand the task's difficulty.**
>
> We were intending to provide a more thorough description along with examples, but regrettably were blocked due to data licensing restrictions. We’re revisiting to see if we can release more details, though in any case we consider the harmlessness and helpfulness tasks to be the more “standard” tasks for our main experiments. We’ll still add some more information on outlining where possible—e.g., the premises are usually 10 to 40 tokens, to get a sense of length. The outputs in our main experiments were anywhere from 20 to 200 tokens or more depending on the method.
>
>
> > **Question 1: How did you decide on the prompt affix pairs?**
>
> Please see description under Concern 1.
>
> > **Question 2: Why is having more than one prompt affix pair important?**
>
> Having more than one pair isn’t required, actually—in our helpfulness task we only use a single prompt affix pair. But we think using multiple affix pairs can be convenient when trying to optimize more complex multi-objective tasks such as outlining, where we are trying to optimize for interestingness as well as well-formedness and premise relevance (hence, we use 3 affix pairs). [1] also used 16 scoring prompts for harmlessness in RLAIF to increase robustness to prompt design, which we matched in our work.
>
> > **Question 3: Given that you automatically assume $o_+$ is preferred - how often does $o_+$ have a lower reward with respect to a held-out reward function?**
>
> We’ve now run this analysis for harmlessness and helpfulness using the same held-out reward models described under Concern 2; see new Appendix K.1. It’s still noisy as we’re relying on the base pretrained LLaMA: the fraction of correct labels w.r.t. the held-out reward ranges from 54% (harmlessness, 7B) to 74% (helpfulness, 30B). However, RLCD’s label accuracy is always higher compared to RLAIF, with the difference ranging from 8% to 14% absolute.

---

> ### Author Response · Authors · 2023-11-17
> **Reply to Reviewer HiRS (part 2)**
>
> > **Question 4: Why would training examples far away from the boundary be better than training examples close to the boundary? I would assume that the points far away could be easy to classify.**
>
> If we could guarantee that the training examples were correctly labeled, then yes we would prefer examples close to the boundary. However, RLAIF/RLCD need to use a model to label the training examples in the first place before they can be used downstream. RLCD’s examples may on average be farther from the boundary, but are more likely to be accurately labeled. In fact, RLCD’s label accuracy may actually be higher *even when considering only examples that are close to the boundary,* as suggested by our theoretical analysis in the new Appendix N; please see “Theoretical Motivation for RLCD” under the General Response.
>
> > **Question 5: Could you elaborate on how you performed your GPT-4 evaluation? What prompt did you use? How did you shuffle the data? Etc?**
>
> The procedure is described in Appendix F.2; the full prompts are shown in Tables 21 and 22. When comparing two outputs, we show them in random order. (Please let us know if you meant something else regarding data shuffling.)
>
> > **Question 6: Could you provide other quantitative metrics for all algorithms considered in your experiments? (e.g., RM-score, perplexity, output-perplexity, etc.)**
>
> Please see Concern 2 above.
>
> > **Question 7: Did you perform GPT-4 evaluation using comparisons from the algorithms-generated output and human-generated data for a given prompt?**
>
> Our GPT-4 pairwise evaluation compares algorithm-generated outputs from two different methods (e.g., RLCD vs. a baseline) on the same prompt. There are no human-generated outputs being compared. (Note we don’t have human-generated *outputs* in the Anthropic data we’re using, only human *preference labels* on model-generated outputs.)
>
> > **Question 8: Could you provide other diversity metrics on the outputs of the text generated? (e.g., the ratio of distinct n-grams (Distinct-1, Distinct-2), average length of sentences, or count of n-grams in the generated text [1]). The text in Table 4 implies that RLCD generations are much longer than the base model sentences.**
>
> Good idea, we’ve now checked these.
>
> The n-gram diversity for RLCD is very similar to that of baselines (new Appendix J.3), except for on harmlessness on 7B scale only, where RLCD’s diversity is lower due to repetitive wording when refusing to answer some requests (though it’s still far from completely mode-collapsed, as shown in examples in Table 26).
>
> As for length (new Appendix J.4), RLCD generations are indeed often longer than those of baselines particularly for helpfulness and story outlining, as RLCD correctly identifies that longer outputs better satisfy these alignment criteria on average (as you noticed in the Table 4 example for the helpfulness task).
>
>
> > **Question 9: Could you elaborate on the RLCD-Resource model setup? In particular, what does it mean to re-label the same scoring prompts as in RLAIF?**
>
> RLAIF: generates two outputs from the same prompt, and then uses a scoring prompt to decide which of the two outputs is better.
>
> RLCD: generates two outputs from different prompts $p_+$ and $p_-$, and then decides which of the two outputs is better simply based on which prompt $p_+$ or $p_-$ was used to generate each output.
>
> RLCD-Rescore: generates two outputs from different prompts $p_+$ and $p_-$ like RLCD, but then decides which of the two outputs is better by using the scoring prompt as in RLAIF, rather than simply based on which prompt $p_+$ or $p_-$ was used to generate each output.
>
> > **Question 10: For RLAIF, did you run an experiment where you sample two outputs from the $p_+$ positive affix prompts? This provides RLAIF algorithms with modified prompts similar to RLCD and would reduce the advantage that RLCD has to strictly have the altered $p_−$ prompts.**
>
> The core intuition of RLCD is to use the different prompts $p_+$ and $p_-$ to amplify the *contrast* between the outputs $o_+$ and $o_-$. In principle, $p_+$ isn’t even needed for contrast—one could imagine a version of RLCD that uses just the base prompt $p$ with a negative prompt $p_-$ to produce $o$ and $o_-$ for contrast instead. So we think that just directly sampling two outputs from the same $p_+$ wouldn’t really change the advantage of RLCD over RLAIF at all, which we confirmed in 7B experiments in the new Appendix L.
>
> [1] Bai, Yuntao, et al. "Constitutional ai: Harmlessness from ai feedback." https://arxiv.org/abs/2212.08073

---

### Author Response · Authors · 2023-11-17
**General Response**

We are grateful to all reviewers for their insightful comments. We appreciate that reviewers found RLCD to be intuitive (HiRS) / clearly-motivated (Cuc6, bkDY), and that it is simple (HiRS, UFW8) / straightforward to implement (RAFU) while still being robust (HiRS) / generalizable (RAFU, bkDY). We also appreciate that all reviewers noted our method’s empirical effectiveness compared to baselines and found our paper to be well-written / easy to follow.

We have now added several new experiments and analyses according to your suggestions, which for now are mainly included in new appendices at the end of the paper for your convenience (so that they’re all in one place, and to keep section numbering consistent with before). We further address each of your questions in the individual responses, in addition to two items which we’d like to highlight here:

**Additional Aspects of Our Contribution**

We would like to draw more attention to the following two aspects of our contribution (and have edited the draft accordingly).

(1) We introduce the idea of modifying not just the process of *labeling* the preference pairs in RLHF / RLAIF, but also the process of *generating* them from the base model in the first place. While it remains to be investigated how well RLCD in its current form scales to models even larger than LLaMA-30B, RLCD is just one possible instantiation of this underlying idea of changing the generation process; we view RLCD as just the first step in this direction. As evidenced by our experiments, changing this generation process can make a substantial difference in practice, so we think that the more general idea of changing the generation process is definitely worth exploring more even for larger model scales. For instance, we suggested some possible modifications to RLCD in our Discussion, and also at the end of our technical analysis in the new Appendix N.

(2) As shown in our experiments, and corroborated by [1], RLAIF performs poorly at small model scales. RLCD provides an alternative approach which works *much* better at 7B model scale, making it easier for researchers and practitioners to try quick experiments on these types of pipelines while spending fewer compute resources.

[1] Bai, Yuntao, et al. "Constitutional ai: Harmlessness from ai feedback." https://arxiv.org/abs/2212.08073


**Theoretical Motivation for RLCD**

We’ve now added a technical explanation for the intuitions behind RLCD.

To summarize: we analyze a simplified setup for preference data simulation where given a prompt, the language model generates responses with attribute value (e.g., harmlessness) distributed according to a Gaussian. Using this setting, we can model RLAIF and RLCD as follows:

* RLAIF first draws two i.i.d. responses from the overall distribution of all possible responses, and then uses a discriminator to label them as positive/negative samples. We assume this discriminator is noisy: specifically, it labels preferences according to the “ground-truth” attribute value of each response, plus some random noise.

* RLCD generates responses from distributions with different means, and labels preferences according to which distributions the responses were sampled from.

As RLAIF needs to label preferences between two i.i.d. samples using a noisy discriminator, RLAIF introduces substantial label noise relative to ground-truth preference labels. Hence RLAIF requires a strong discriminator (e.g., 30B or even larger LLM) to start functioning well. On the other hand, RLCD does not suffer from these issues and can tune the attribute values of positive/negative samples by changing the prompts accordingly.

In fact, we find that not only can RLCD achieve much more accurate preference labels overall compared to RLAIF by making responses more contrastive, but in doing so RLCD can also achieve a higher accuracy of preference labels *even when considering only “hard” examples close to the boundary,* which we verify in simple simulations.

Please refer to the new Appendix N for the complete details of the setup, analysis, and quantitative results.

---

> ### Author Response · Authors · 2023-11-21
> **Please let us know if you have any further questions!**
>
> We thank all reviewers again for their insightful comments and questions.
>
> We have attempted to carefully address all of your questions in the individual responses and in the edited paper draft, and would be happy to follow up if you have any further questions in the last two days of the discussion period.

---

### Meta-Review · Area_Chair_RXnX · 2023-12-06

**Metareview:**

This paper introduces RLCD (Reinforcement Learning from Contrastive Distillation), a novel method for aligning language models with principles expressed in natural language. RLCD generates preference pairs using contrasting prompts to create preference labels without human feedback, and then use these pairs to train a preference model through reinforcement learning for enhancing  an unaligned language model. The paper demonstrates RLCD's superior performance over existing methods like RLAIF in context distillation across various tasks and at different model scales.

Reviewers acknowledged the intuitive and simple yet effective nature of RLCD, highlighting its empirical strengths and potential value to the community. They appreciated the paper's clarity and the method's ease of implementation. However, concerns were raised about RLCD's performance attribution and the simplicity of the approach. Reviewers also inquired about the method's scalability and sought more technical details, particularly regarding the fine-tuning procedures. In response, the authors conducted additional experiments and analyses, providing theoretical motivations and detailed explanations of the PPO fine-tuning process. These responses effectively addressed the reviewers' concerns.

Considering the novelty, empirical strength, and potential impact of RLCD, alongside the authors' comprehensive responses to reviewer concerns, the paper is accepted. The authors have convincingly shown a method on AI self-alignment without heavy reliance on human feedback.

**Justification For Why Not Higher Score:**

Concerns about its scalability and mixed evaluation results suggest that the work, though important, may not fully align with a higher level of ICLR’s  recognition.

**Justification For Why Not Lower Score:**

The authors have convincingly shown a method on AI self-alignment without heavy reliance on human feedback.

---

### Decision · Program_Chairs · 2024-01-16

Accept (poster)